# Structural basis of bacterial effector protein azurin targeting tumor suppressor p53 and inhibiting its ubiquitination

Jianjian Hu[1,3], Wenxue Jiang[2,3], Jiaqi Zuo[1,3], Dujuan Shi[2,3], Xiaoqi Chen[2], Xiao Yang[2], Wenhui Zhang[1], Lixin Ma[2], Zhu Liu [2] & Qiong Xing [2✉]

Tumor suppressor p53 prevents tumorigenesis by promoting cell cycle arrest and apoptosis through transcriptional regulation. Dysfunction of p53 occurs frequently in human cancers. Thus, p53 becomes one of the most promising targets for anticancer treatment. A bacterial effector protein azurin triggers tumor suppression by stabilizing p53 and elevating its basal level. However, the structural and mechanistic basis of azurin-mediated tumor suppression remains elusive. Here we report the atomic details of azurin-mediated p53 stabilization by combining X-ray crystallography with nuclear magnetic resonance. Structural and mutagenic analysis reveals that the p28 region of azurin, which corresponds to a therapeutic peptide, significantly contributes to p53 binding. This binding stabilizes p53 by disrupting COP1-mediated p53 ubiquitination and degradation. Using the structure-based design, we obtain several affinity-enhancing mutants that enable amplifying the effect of azurin-induced apoptosis. Our findings highlight how the structure of the azurin-p53 complex can be leveraged to design azurin derivatives for cancer therapy.

[1] National Key Laboratory of Crop Genetic Improvement, Hubei Hongshan Laboratory, Huazhong Agricultural University, Wuhan 430070, China. [2] State Key Laboratory of Biocatalysis and Enzyme Engineering, College of Life Sciences, Hubei University, Wuhan 430074, China. [3] These authors contributed equally: Jianjian Hu, Wenxue Jiang, Jiaqi Zuo, Dujuan Shi. ✉email: qiongxingnmr@hubu.edu.cn

The tumor protein p53 has well-characterized functions in regulating cell survival and death under certain stimuli, including bacterial infection[1]. It is a transcription factor that activates genes to induce apoptosis, senescence, and cell cycle arrest[2]. The downstream effectors of p53 include the apoptosis regulator Bcl-2-associated X protein Bax and the cyclin-dependent kinase inhibitor protein p21[3]. Frequent inactivation of p53 in human malignancies renders it a highly attractive target for new anti-cancer drugs[3,4]. Human p53 protein assembles into a dynamic homo-tetramer and harbors three major functional domains: the N-terminal domain (NTD, residues 1-94) including a transcriptional activation domain (TAD, residues 1-61) and a proline-rich domain (PRD, residues 62-94), the DNA-binding domain (DBD, residues 95-292), and the C-terminal domain (CTD, residues 293-390) including tetramerization domain (TD, residues 293-353) and regulatory domain (RD, residues 354-390) (Fig. 1a)[4]. Normally, the three individual domains are in charge of three fundamental events of p53 activation, p53-CTD for structural and functional stabilization of p53, p53-DBD for sequence-specific recognition and binding of DNA, and p53-NTD for transcriptional initiation of target genes[5].

Due to the critical function of determining cell rate, the lifetime and basal level of p53 are highly regulated[6,7]. This regulation is primarily monitored by ubiquitin-mediated protein degradation[5,8]. In cells, intact p53 is directly targeted by ubiquitin E3 ligases such as Mdm2 or COP1 for ubiquitination and pro-teasomal degradation[9]. When exposed to extreme condition, Mdm2 or COP1 is universally modified and inhibited, and con-sequently, lead to continuous activation of p53[9]. However, COP1 or Mdm2 is commonly overexpressed in various cancers, which leads to accelerated degradation of p53 and attenuation of p53-induced apoptosis[10,11]. Therefore, targeting on E3 ligase-mediated p53 ubiquitination pathway to retain the function of p53 is an attractive strategy for the treatment of specific cancers[12].

Increasing evidence highlights the potential of proteins or peptides from pathogenic bacterial microbiota targeting p53 stabilization in cancer therapy[1,13–17]. Among the reported candidates, azurin can be regarded as one of the most important and well-studied proteins18. It is a 128-amino acid redox protein secreted by an opportunistic pathogen, *Pseudomonas aeruginosa*, which consists of an immunoglobulin fold and an α-helix protein transduction region (L50–D77, p28) (Fig. 1a)[18]. This effector

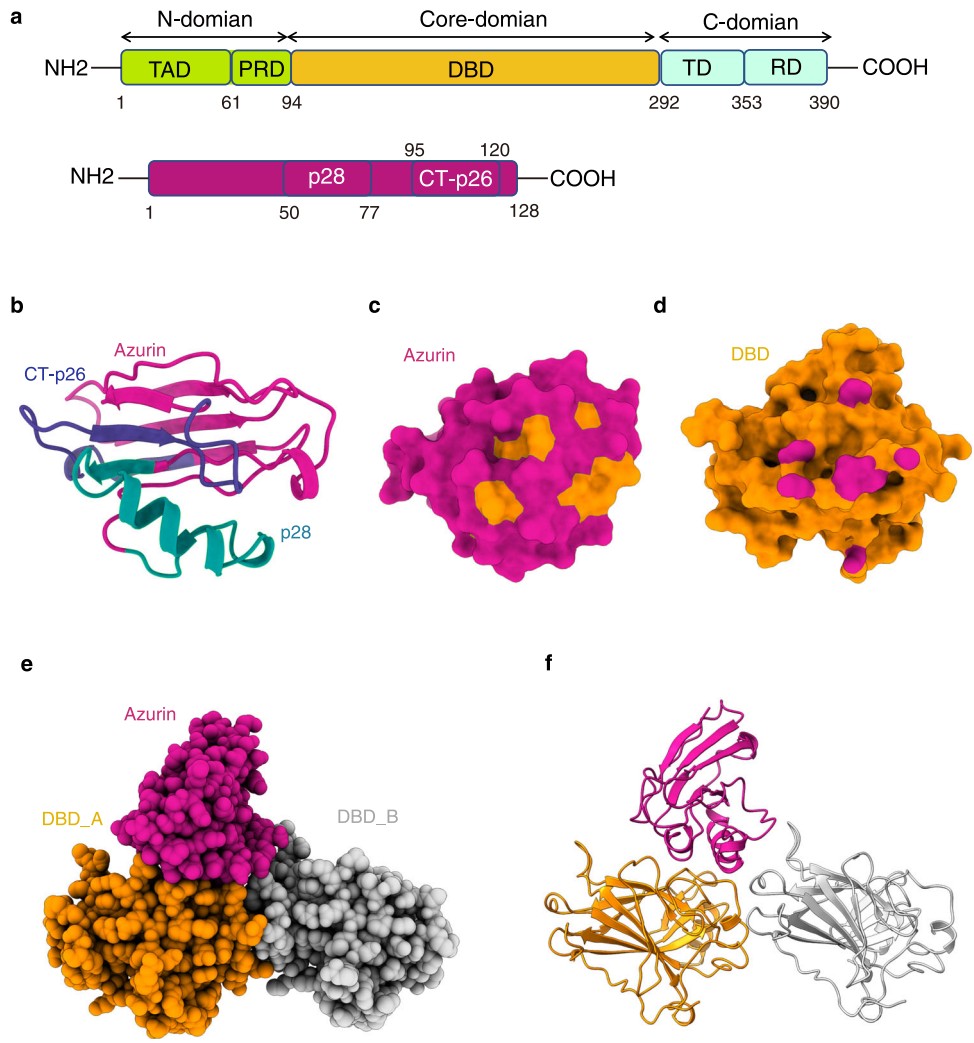

**Fig. 1 Characterization of the intermolecular interactions and structure between azurin and p53. a** Domain organization of human p53 and *pseudomonas aeruginosa* azurin. **b** Structure of azurin shown as cartoon (colored medium violet). Azurin p28 (colored dark cyan) and CT-p26 regions (colored marine) are highlighted. **c** The site of azurin interacting with p53 determined by solution NMR. Key residues with chemical shift perturbations are colored in orange. Azurin is rendered as surface colored medium violet. **d** The azurin binding site on p53-DBD (colored medium violet) determined by NMR titration. p53-DBD is rendered as surface and colored in orange. **e** Crystal structure of the azurin in complex with p53-DBD shown as a spacing-filling style. The two p53-DBD molecules are colored in orange and silver, and one azurin molecule is colored in medium violet. **f** Cartoon representation of the azurin-p53 structure.

protein has high selectivity and cytotoxicity to a variety of human cancer cells, including breast cancer and melonoma[19–22], which makes azurin and its derivatives an alternative route to cure cancer[23]. Currently, two regions of azurin have been explored as the hotspots for membranolytic anti-cancer peptide development, the p28 region and CT-p26 region (V95-L120), respectively (Fig. 1a, b)[18,24]. Notably, the azurin-derived peptide, p28 (L50-D77 of azurin), has been approved by FDA as an orphan medication for treating glioma[25] and validated the safety and antic-ancer activity in adult as well as pediatric patients[14,26].

Numerous studies have shown that azurin and its derived peptides preferentially penetrate tumor cells and trigger apoptosis by stabilizing and activating p53[19,20]. The lack of structural and mechanistic information on the p53-azurin complex has limited our understanding of azurin-regulated p53 stabilization and impeded the development of new anticancer azurin-derived peptides.

Here, we report the 2.1 Å crystal structure of azurin in complex with human p53 and provide atomic insights into how azurin stabilizes p53. Our findings show that this transient azurin-p53 interaction inhibits COP1-mediated p53 ubiquitination and leads to intracellular p53 accumulation. Indeed, designed mutations in azurin favoring p53 binding upregulate protein levels of p53 as well as its transcriptional targets that trigger apoptosis, including the protein p21 and Bcl-2-associated X protein Bax. Our findings reveal the molecular mechanism of azurin-induced inhibition of p53 ubiquitination.

## Results

**Structure of azurin in complex with p53.** To probe the binding mechanism of p53-azurin, we used solution nuclear magnetic resonance spectroscopy (NMR) and titrated isotopically labeled azurin with full-length p53 as well as its domains. NMR data analysis indicated that full-length p53 and p53-DBD domain bind to the same region of azurin, while no significant interaction was observed between azurin with either the p53-NTD or p53-CTD domain of p53 (Supplementary Fig. 1a, b, c, and d). Most per-turbed residues form a contiguous surface on azurin (Fig. 1c). Among the most perturbed residues, including G63, M64, D69, Y72, D76, and D77, are dispersed in the p28 region, and G116 and S118 are located at the CT-p26 region (Fig. 1b, c). These results suggest that the reported anti-cancer peptides derived from CT-p26 region[24], gain cytotoxic activity by targeting p53 similar to p28 and p28-derived peptides.

Analyses of the NMR titration data obtained by adding unlabeled azurin to $^{15}N$ labeled p53-DBD indicated that two loop regions, S183-L188 and V197-L201 (Fig. 1d, Supplementary Fig. 1e, f), contribute to the recognition of azurin. We note that chemical shift perturbations can only be observed with unlabeled azurin at concentrations up to 1.4 mM titrated into 0.4 mM $^{15}N$-labeled DBD, suggesting a weak binding affinity (Supple-mentary Fig. 1e, f). Size exclusion chromatography and SDS page analysis reveals that azurin, p53 and p53-DBD are highly homogeneous (Supplementary Fig. 1g, h, i), and azurin barely comigrated with p53-DBD (Supplementary Fig. 1j, k), indicating that the interaction between azurin and DBD is transient and dynamic, in agreement with the NMR titration results.

To understand how azurin binds to and stabilizes p53, we sought to determine the structure of azurin in complex with p53. The initial trial was to crystallize the p53-azurin complex by incubating azurin with full-length p53 or p53-DBD in high concentrations. Nevertheless, no crystal was obtained most likely due to unfavorable kinetics or low population of protein complex in solution. To stabilize the DBD-azurin complex, we turned to the fusion strategy. Fusion proteins mimic proteins in extremely

**Table 1 Data collection and refinement statistics.**

|  | P53_DBD-Azurin |
|---|---|
| Data collection | |
| Wavelength | 0.979 |
| Resolution range (Å) | 19.88 – 2.1 |
| Space group | C 1 2 1 |
| Unit cell | |
| a, b, c (Å) | 144.84 68.75 83.85 |
| α, β, γ (°) | 90.00 99.70 90.00 |
| Total reflections | 222394 (15529) |
| Unique reflections | 46000 (4687) |
| Multiplicity | 4.9 (5.0) |
| Completeness (%) | 96.18 (99.22) |
| I/σ(I) | 7.7 (3.1) |
| R-merge (%) | 14.0 (42.9) |
| R-meas (%) | 15.6 (48.0) |
| R-pim (%) | 6.8 (20.9) |
| CC$_{1/2}$ | 0.989 (0.908) |
| Refinement | |
| Reflections | 45781 (4684) |
| Reflections used for R-free | 2256 (235) |
| R-work | 21.3 (28.6) |
| R-free | 26.9 (34.7) |
| Number of non-hydrogen atoms | 5688 |
| macromolecules | 4967 |
| ligands | 11 |
| solvent | 710 |
| Protein residues | 642 |
| r.m.s. deviation | |
| Bond lengths (Å) | 0.009 |
| Bond angles (°) | 1.28 |
| Ramachandran plot | |
| Favored (%) | 97.00 |
| Allowed (%) | 3.00 |
| Outliers (%) | 0.00 |
| Average B-factor | 31.48 |
| macromolecules | 31.27 |
| ligands | 46.98 |
| solvent | 32.66 |
| PDB ID | 7YGI |

Statistics for the highest-resolution shell are shown in parentheses.

high concentrations and dramatically elevate the population of transient protein complexes in solution. This approach has been extensively used in the structure determination of transient protein complex by crystallography and solution NMR[27,28]. Based on the models from HADDOCK[29] server supplied with restraints from NMR titrations (Supplementary Fig. 2), several fusion constructs were designed and tested. The highest quality crystals suitable for structural determination were given by a construct of fusing azurin to the C-terminal of p53-DBD with a glycine-serine repeats linker (GGSGSGSLVPRGGSGGSGGS) (Supplementary Fig. 3a). This protein construct was used for further crystal optimization and X-ray crystallographic structural determination. Eventually, the structure of p53-DBD in complex with azurin was determined at 2.1 Å resolution (Fig. 1e, Table 1).

In the structure, each crystal unit cell contains two azurin and two p53-DBD molecules. Two p53-DBD molecules dimerize in an asymmetric way and form a triangle groove (Fig. 1f, Supplementary Fig. 3b). An azurin molecule (azurin_A) docks into the groove. To confirm the dimerization of p53-DBD in p53-azurin complex, we carried out sedimentation velocity analytical ultracentrifugation (SV-AUC) experiments, which showed that the fusion protein is in an equilibrium of monomer and dimer in solution (Supplementary Fig. 3c) and confirmed that the dimerization of p53-DBD is not due to crystal packing.

Importantly, no physical interaction was observed between azurin_B and p53-DBD (Supplementary Fig. 3b). In the crystal packing diagram for this complex, the C-terminal of azurin_B is too far away from the N-terminal of p53-DBD_B and very near to p53-DBD_B in the adjacent crystal unit (Supplementary Fig. 3d), which suggest that the azurin_B molecule present in the unit cell is a result of fusion-induced crystal packing.

**The interface of the azurin-p53 complex.** The structure of the azurin-p53 complex is shown in Fig. 1f. The complex is featured with a distinct azurin-p53 interface between one azurin molecule and one p53-DBD molecule. Overall, the ternary complex is stabilized by hydrophobic interaction as well as electrostatic interaction (Supplementary Fig. 4a). Dimerization of p53-DBD forms a triangle groove, which is highly complementary to the triangle shape of azurin (Supplementary Fig. 4b). This indicates the contribution of van der Waals interactions in the complex formation. The interface is mainly mediated by the p28 region, burying a total surface of 913 Å$^2$ (Fig. 1f). This interaction mode is consistent with the information we obtained from NMR titration (Supplementary Fig. 1c, e). In this configuration, two bulky hydrophobic residues from p53, Leu188, and Leu201, stick out their side chains to interact with a hydrophobic pocket of azurin consisting of three non-polar residues, L39, V43, and M44 (Fig. 2a). In addition to the intimate non-polar contacts, the binding interface features two critical intermolecular hydrogen bonds, which are formed between backbone α-carbonyl of azurin Ser66 and the side chain of p53 S183 (O–H···O distance 3.0 Å), and between the side chain of azurin Y72 and backbone α-carbonyl of p53 S185 (O–H···O distance 1.8 Å) (Fig. 2a). Besides these key interactions, several residues at the interface involved in intermolecular van der Waals interactions also contribute to azurin-p53 binding (Fig. 2a).

Substitutions of amino acid located at the binding interface decreased the stability of the p53-azurin complex, especially for mutations perturbing the optimal hydrogen bond formation (Fig. 2b). All of these residues are highly conserved through 20 bacteria strains (Supplementary Fig. 5). An oncogenic mutation of p53, S241F, leads to antitumor resistance of azurin[30]. Interestingly, S241 is located at the DBD dimer interface and involved in p53-DBD dimerization by forming two intermolecular hydrogen bonds (Fig. 2a). Changing this small residue to an aromatic residue with bulky side chain would destabilize the p53-DBD dimer. Our data showed that this mutation disrupts the azurin-p53 interaction (Fig. 2b), indicating that p53-DBD dimerization plays a role in azurin binding.

In complementary to the single point mutation analyses, we performed chemical cross-linking-coupled mass spectrometry to validate the relative configuration of azurin docking on p53-DBD in solution (Supplementary Fig. 6a). Four distinct intermolecular lysine pairs, p53 K120-azurin K74, p53 K120-azurin K101, p53 K120-azurin K103, and p53 K120-azurin K128, were crosslinked and identified (Supplementary Fig. 6b). While no intermolecular lysine pairs from DBD dimer was identified due to distance limit of crosslinking reagent (Supplementary Fig. 6b). These results are highly consistent to our structure (Fig. 2c).

**The molecular mechanism underpinning p53 stabilization by azurin.** Based on our structural and pull-down data, several residues were identified as critical for the complex formation (Fig. 2c). Among them, most of the residues, including Q57, T61, M64, S66, Y72, are located at azurin p28 region (Fig. 3a). The hydrophobic and hydrophilic nature of these residues are highly conserved in >20 species of bacteria strains (Fig. 3b). Notably, p28 peptide, an anti-glioma drug, specifically dampens the COP1-p53

binding and stabilizes p53 from degradation[31], and our structural analyses showed that there are overlaps between azurin binding sites and reported ubiquitin E3 ligase COP1 binding sites on p53[14,32]. suggesting that azurin might negatively regulate COP1-mediated p53 ubiquitination through competitively binding to p53.

To illustrate the biological relevance of the azurin-p53 interaction, reported herein, we sought to investigate the effects of azurin on COP1-mediated p53 ubiquitination. Using in vitro protein ubiquitination assay, we established that p53 was readily ubiquitinated by COP1 in cooperation with E1 and E2 (Fig. 3c). Intact azurin was capable of disrupting the COP1-mediated p53 polyubiquitination, in a concentration-dependent manner (Fig. 3c). We evaluated p53 ubiquitination by COP1 with each azurin mutant. Notably, single amino acid substitution on azurin, which ablates the complex stability, markedly restored the ubiquitination level of p53 (Fig. 3d). Previous reports[21] showed that mutation at azurin M44 and M64, which are located exactly at the p53-azurin interface (Figs. 2a, 3a), have pronounced defect in p53 stabilization and cytotoxicity, in agreement with the present structural and functional data, highlighting the great importance of these residues. The way the p28 peptide fulfills its function is highly consistent with azurin, indicating that p28 binds to p53 in the same way as azurin p28 region. Then, the structure is of great importance for p28 peptide refinement and azurin-based peptide design with high anticancer activity. Hence, the binding interface reported here is crucial to the biological function and clinical significance of azurin.

**Structural-guided design of affinity-enhancing azurin mutant.** The transient and dynamic nature of p53-azurin interaction has been the major hurdle to improve the anti-cancer efficacy and specificity of azurin and its derived peptides. The structure of p53-azurin complex reported here enables to guide structure-based mutant design to enhance the interaction. Owing to the high resolution, all the intermolecular contacts can be visualized in details (Supplementary Fig. 7). In fact, the structure reveals four distinct regions consisting of residue pairs from p53 and azurin that are juxtaposed within a close distance <4 Å (Fig. 4a). The criterion for designing mutants exhibiting enhanced binding affinity is to maximize the formation of salt bridge, hydrogen bond or hydrophobic interaction between intermolecular amino acids. Of note, azurin S66 and Y72 are directly involved in intermolecular hydrogen bond formation and make a major contribution to the binding. Thus, mutation design should keep a distance from these two residues. Accordingly, we designed a series of azurin mutants that potentially stabilize the complex, and tested their alterations in p53 binding affinity using immunoprecipitation assays. The data showed that azurin mutations, including M13I, V43I, Q57E, T61D, S78I, G116K and A119K, led to notable increases in binding affinity towards p53 (Fig. 4b). The effect of designed azurin mutations on p53 ubiquitination, mediated by COP1, was assessed by ubiquitination assay (Fig. 4c). The mutants Q57E and S78I, potentially forming a salt bridge and non-polar interaction with p53 K139 and V225 respectively (Fig. 4a), significantly decreased the ubiquitination level of p53 (Fig. 4c). Another set of azurin mutants, V43I and A119K, which enhance the hydrophobic and electrostatic interaction with p53 L188 and E198 respectively, also reduced p53 ubiquitination (Fig. 4c). Collectively, these structure-guided mutagenesis assays provided mutations of azurin with significant affinity enhancement towards p53 and affirmed our structural as well as functional results.

**Affinity-enhancing mutations promote p53-regulated apoptosis.** The stability, activity and dynamics of p53 in cell varies depending on the external stimulations[1]. Azurin and its derived

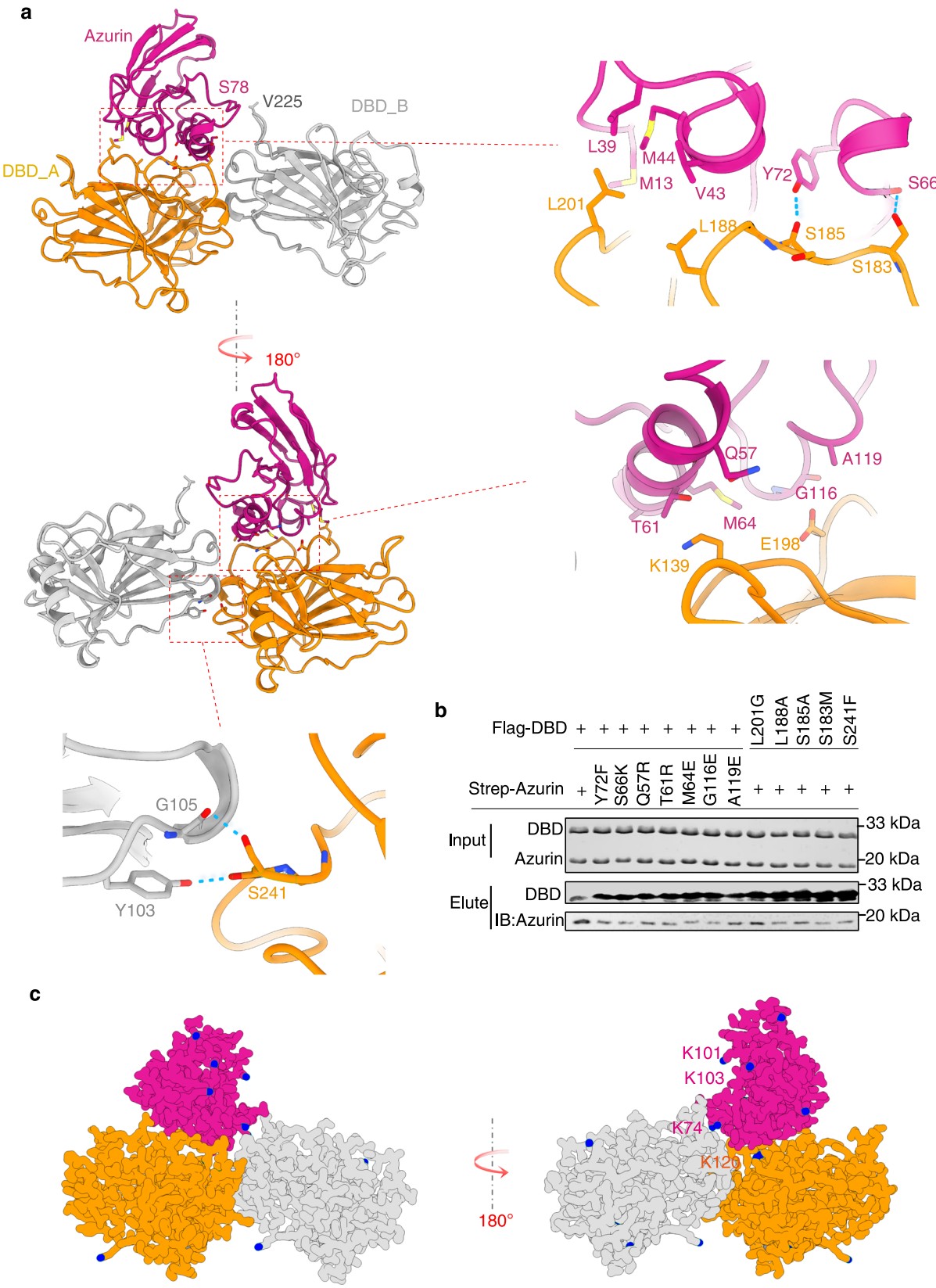

peptide treatment is one of the stimulations. To delineate the clinical potentials of the designed azurin affinity-enhancing mutations in vivo, we examined the effect of azurin as well as its mutant treatment on p53 mediated apoptosis using Annexin V/PI staining and flow cytometry. Our data showed that HEK 293 T cells treated with phosphate buffer for 48 h have about 3.2%

of apoptotic cells. While cells treated with azurin WT, azurin V43I, azurin Q57E, azurin S78I and azurin A119K, led to ~7.3%, 9.0%, 10.4% and 12.7% of cell death, respectively (Fig. 5a, b).

Additionally, the level of p53 was marginally elevated with WT azurin treatment as compared to phosphate buffer treatment (Fig. 5c). However, when cells treated with azurin mutants that

**Fig. 2 Structural features of azurin in complex with p53. a** Cartoon rendering of the azurin-p53 structure and expanded views of the azurin-p53 interface is highlighted. Residues contributing to intermolecular contacts are presented as ball-and-stick. Intermolecular hydrogen bonds are shown as broken lines colored in cyan. **b** Immunoblots assay showing the effect of point mutations on the binding affinity of azurin and p53-DBD. This assay was performed with anti-Flag beads which specifically binds to Flag-DBD or its mutants to pull down strep-azurin in solution, and the elution was blotted with anti-flag and anti-strep antibodies, respectively. The intensity of the blot of azurin represents the relative affinity of the azurin and p53-DBD. **c** Chemical crosslinking mass spectrometry analysis of the azurin-p53 complex. The crosslinking reagents used are specifically react with lysine. All the Nz atom of lysine residues are shown and highlighted in blue. The crosslinked pairs captured are connected with yellow broken lines.

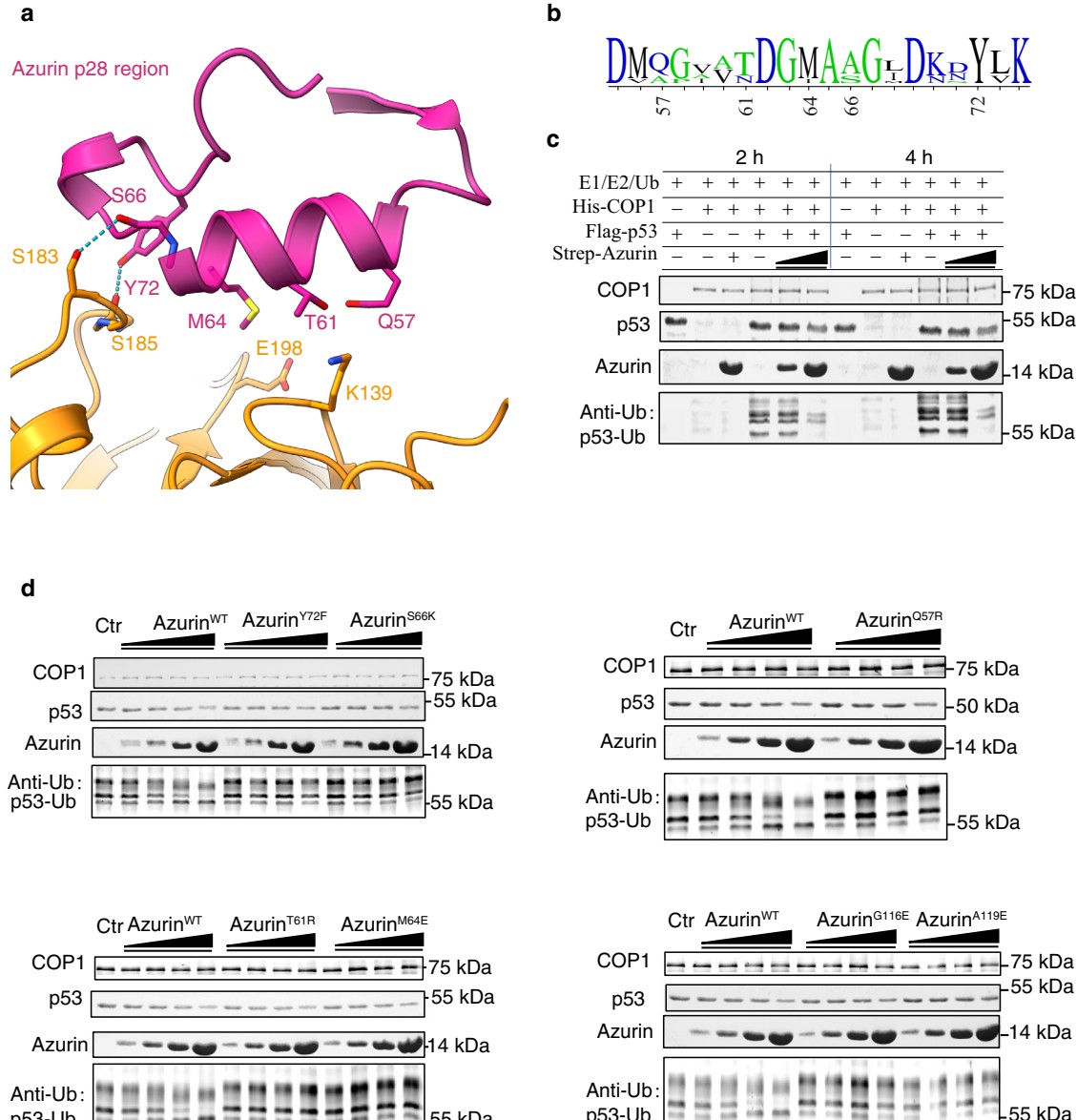

**Fig. 3 Dissection of the effect of azurin on p53 ubiquitination. a** Highlighting the detailed interaction of azurin p28 region and p53-DBD. Residues involved in binding are labeled and shown as sticks. **b** Sequence conservation of residues at azurin p28 region involved in p53 binding. **c** In vitro ubiquitination assay of COP1-mediated p53 ubiquitination intervened by azurin. The reaction mixtures were incubated for 2 and 4 h, and then resolved by SDS-PAGE and immunoblotting assay with exclusive antibodies. **d** Effect of the affinity-disrupting mutants on ubiquitination level of p53 measured by ubiquitination assay. The designed mutants are azurin Q57R, T61R, M64E, G116E and A119E. Ctr represents p53 ubiquitination without any azurin treatment.

increased the p53-azurin complex stability, p53 levels elevated dramatically (Fig. 5c). In contrast, treatment with azurin mutants introduced to amino acids locating beyond the p53-azurin interface, had no appreciable impact on the p53 levels (Supplementary Fig. 8). These results reaffirmed that azurin functions to govern p53 level in cells and that the affinity-enhancing mutations amplify this effect. It has been shown that

p53 induces apoptosis through regulating the transcription levels of downstream effectors, which initiate the onset of apoptosis, including Bcl-2-associated X protein Bax and protein p21[33]. To determine if azurin-induced apoptosis is associated with p53-dependent transcriptional regulation of these proteins, we sought to use immunoblotting assay to characterize the in vivo induction of p21 and Bax. Our data demonstrated that cells

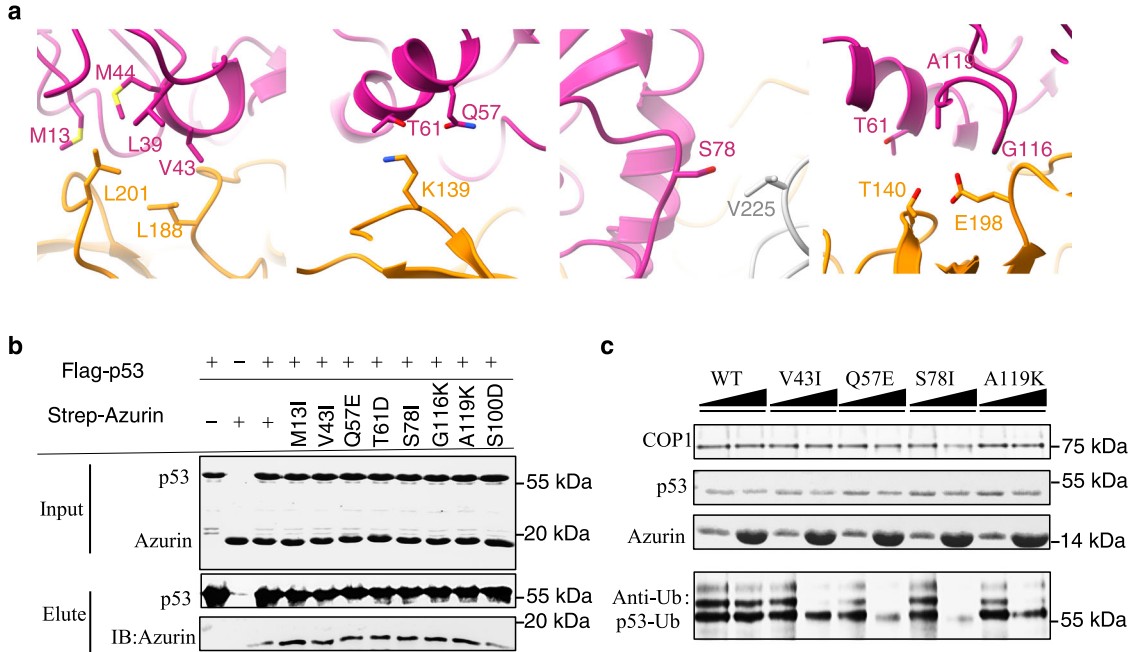

**Fig. 4 Structural-guided-design of affinity-enhancing mutations on azurin. a** Cartoon and stick rendering for residues involved in intermolecular contacts of azurin-p53 complex within a distance <4 Å. **b** Pull down assay validating the effect of designed affinity-enhancing mutations on the binding affinity of azurin-p53 complex. Anti-flag beads were used to specifically binds flag tagged p53 to pull down strep tagged azurin or its mutants in solution. Semi-quantification of azurin or p53 in elution was measured by western-blot with anti-flag and anti-strep antibodies. Azurin S100 is located far away from the interface and the S100D mutation here is served as negative control. **c** Effect of the affinity-enhancing mutants on the ubiquitination level of p53. The azurin mutants are V43I, Q57E, S78I and A119K.

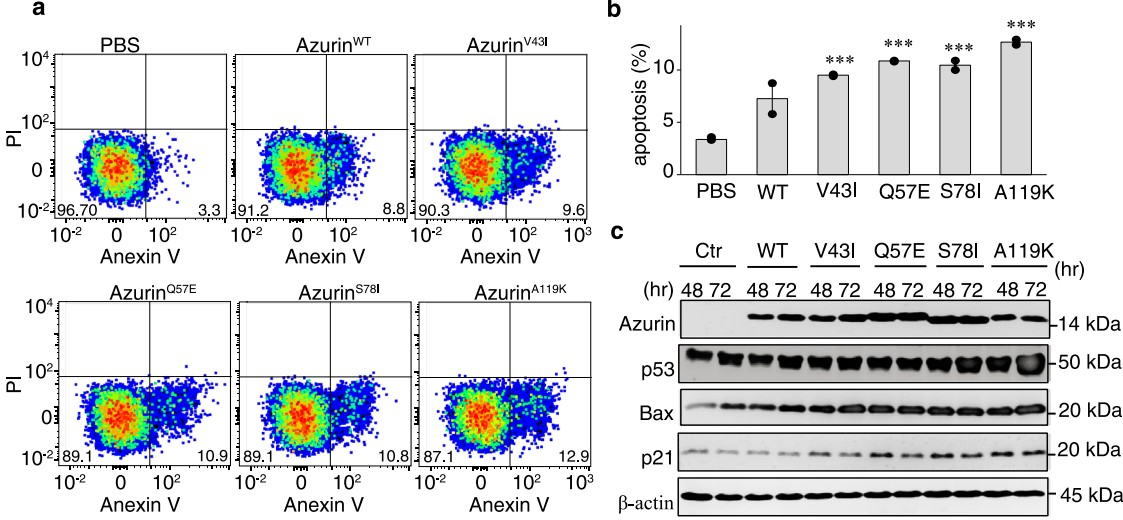

**Fig. 5 Affinity-enhancing mutations promote p53-regulated apoptosis. a** Flow cytometry profiles of cells treated with different type of reagents, containing phosphate buffer as control, azurin and its four mutants. **b** Statistics of the apoptosis rate of cells treated with azurin or its mutants obtained from three paralleled experiments. Data are presented as dot, $n = 3$ independent replicates. Error bars indicate mean ± SEM. *P*-values were determined by one way ANOVA followed by Dunnett's multiple comparisons test using PBS group as a control, **$P < 0.001$, ***$P < 0.0001$. **c** Immunoblotting analysis of protein level in cells treated with azurin and the affinity-enhancing mutants. The antibodies for each protein are anti-his antibody for azurin, anti-p53 antibody for p53, anti-p21 antibody for p21, anti-BAX antibody for BAX and anti-beta-actin antibody for beta-actin.

treated with azurin showed slightly increase in Bax and p21, and both of them were increased significantly in cells treated with affinity-enhanced azurin mutants (Fig. 5c).

## Discussion
The tumor suppressor p53 functions to regulate cell death and cell proliferation under certain stress. Its activation leads to apoptosis, senescence and cell cycle arrest[6,7]. In fact, p53 mutations occurs in 50% of human cancers and results in continual unregulated proliferation of tumor cells[34]. Thus, restoring p53 function in human cancers is a promising therapeutic approach. The bacterial effector protein azurin specifically binds to p53 and triggers tumor suppression by elevating the basal level of p53[19]. While the structural basis of how azurin mediates p53 stabilization and dynamics (p53 level versus time) had been

hitherto elusive. The present data reveal the structural features underlying the stabilization mechanisms of tumor suppressor p53 by a bacterial effector protein azurin.

Noteworthy, the azurin-p53 binding cleft in p53 is located at the p53 DBD domain. We compared the azurin-p53 structure reported here with the structure of p53 in complex with a host effector protein BCL-xL determined earlier. Interestingly, although the structural basis in detail is totally different, azurin binds to the similar area of p53 dimer as anti-apoptotic protein BCL-xL[28] (Supplementary Fig. 9a). These comparisons highlight the possibility that azurin competes with BCL-xL for p53 binding and then resumes the BCL-xL protein induced anti-apoptosis processes, a meaningful hypothesis that needs to be further validated in future studies.

The stoichiometry of the p53-azurin complex we obtained from our crystal structure is 1:2, which is quite different from the previous reported 1:1 stoichiometry. Firstly, our SV-AUC data confirmed the dimerization of p53-DBD in azurin-p53 complex. Secondly, the dimer interface of p53 DBD reported here is highly identical to previous reported p53 DBD structures[28,35], which are stabilized by the same intermolecular hydrogen bond network coordinated by Y107, H178 and D259 (Supplementary Fig. 9b). Last but not least, an oncogenic p53 mutant, S241F, proves to be insensitive to azurin-induced apoptosis[30]. This mutation is located at the p53 DBD dimer interface but not p53-azurin interface, and still enables to disrupt the interaction and restores the effect of azurin treatment. All of the evidence backup the concept that the dimerization of p53 is important for the p53-azurin complex stabilization. Stable tetramerization of p53 in cell would enhance the interaction for azurin, which indicates the effect of azurin induced p53 stabilization would be amplified in cell comparing to our in vitro ubiquitination assay.

The key interactions appear to be two intermolecular hydrogen bonds. Residues on azurin involved in the key interactions are S66 and Y72. Of note, both residues are located in the p28 region and appear to be highly conserved, which indicates the crucial importance of these two amino acids to engage azurin-p53 interaction. Importantly, the solution structure of p28 peptide appears to be highly identical to azurin p28 region[36], indicating that the structure reported here enables to reflect the interaction between p28 peptide and p53. Therefore, drug design based on refinement of p28 sequence needs to pay close attention to these two residues.

In addition to the key interactions, several residues from the azurin CT-p26 region (residues 95–120) or CT-p19 region (residues 99–117) also contribute to the interaction (Figs. 2b, 3a). Previous reported work supports that azurin derived peptides, CT-p26 and CT-p19, also present great propensity for anticancer activity[24]. This result suggests that these two peptides probably gain cytotoxic activity by targeting on p53 in the same manner as p28. Besides, the structure of CT-p26 and CT-P19 region in azurin is largely random coil, which indicates that peptide derived from these two regions is also disordered. If this hypothesis could be validated, the structure reported here would be of great importance for the development of new bioactive peptides against tumor cells.

Currently, major efforts are devoted to develop drugs to restore p53 stability and activity[37–39]. Using structural-guided mutagenesis based on our result, we obtained several affinity-enhancing variants of azurin. These azurin variants enable to stabilize the azurin-p53 complex and significantly restore the p53 stabilization from ubiquitination-mediated degradation. Specifically, the designed mutations as Q57E and S78I are related to p28 region, and A119K is involved in the CT-p26 peptide. In this regard, these mutations can be transferred directly to p28 peptide or CT-p26 peptides. These results

provide great potential to dramatically improve the antitumor activity of these peptides.

Taking together, the present data provide atomic insight into the structural features underlying the mechanisms of p53 accumulation induced by a bacteria effector azurin. Azurin exposure leads to apoptotic cell death through stabilizing p53 from COP1-mediated ubiquitination and proteasomal degradation. Our structure-guided mutagenesis assays affirmed the structural and functional results, and provided perspectives for designing azurin mutants with greater anti-tumor potentials.

## Methods

**Gene cloning, protein expression and purification.** The DNA sequences encoding *Pseudomonas aeruginosa* Azurin, excluding its N-terminal 20 residues which are predicted to be signal peptide, was inserted into pET-15d vector with a N-terminal Strep tag. Human p53 full-length and its individual domains (Fig. 1A) were cloned into pET-15d vector with a N-terminal 6×His-tag and a C-terminal 3XFlag tag (DYKDDDDK). Plasmids containing His-GST-UBE1, His-UBcH5b, His-hCOP1 or MSAC-Ub, are precious gifts from Prof. Ping Yin's lab. For crystallization, a series of fusion constructs coding p53-DBD and azurin were designed and made by fusing with two proteins with a Glycine-Serine rich linker (GGSGSGSLVPRGGSGGSGGS). Primer information related to making all the constructs are summarized in Supplementary Table 1. For protein expression, BL21 (DE3) Cells were induced with 0.4 mM isopropyl β-D-1-thiogalactopyranoside (IPTG) and 0.2 mM ZnCl$_2$ at an optical density at 600 nm around 0.8 and grown at 16 °C overnight. p53 full-length and related individual domains as well as the fusion constructs were purified by nickel affinity chromatography and Heparin HP (HP×5 ML, GE Healthcare). Further purification was conducted by size exclusion with a Superdex 200 10/300 (GE Healthcare) column, pre-equilibrated with buffer containing 25 mM Tris-HCl (pH 8.0), 200 mM NaCl, 5 mM DTT. Azurin and its mutants were purified by affinity chromatography with Strep-Tactin Sepharose column (IBA Lifesciences). Further purification was performed with size exclusion using the same buffer as p53 protein. UBE1, UbcH5b, COP1 and MSAC-Ub (Four residues Met, Ser, Ala and Cys appended at the N-terminus of ubiquitin)[40] were purified by nickel affinity chromatography, and followed by size exclusion with the same condition as p53. Peak fractions were collected and stored at −80 °C. All mutant proteins were purified by the same methods.

**Protein isotope labeling and solution NMR spectroscopy.** $^{15}$N labeled protein samples were expressed in BL21(DE3) cells and grown in minimal medium (M9) supplied with 1 g/L $^{15}$N-NH$_4$Cl. Cells were induced at around OD600 = 0.9 with 0.5 mM IPTG at 23 °C for about 16 h. The harvested cells were stored at −20 °C. Purification procedure for isotope labeled proteins is the same as unlabeled protein samples. The $^{15}$N-NH$_4$Cl were purchased from Cambridge Isotope Laboratories and Isotec. The NMR titration experiments were carried out on Bruker AVANCE III 800 MHz instruments equipped with cryogenic probes. All the samples for NMR experiments were prepared in 25 mM NaPi (pH 7.0), 150 mM NaCl, 5 mM DTT and 10% deuterated D$_2$O. Concentration of isotopically labeled sample, azurin or p53-DBD, was kept the same as 0.5 mM. Unlabeled proteins were prepared at 1.2 mM, except for full-length p53 at 0.3 mM. The $^1$H-$^{15}$N-HSQC spectra were recorded at 25 °C and processed with NMRPipe[41] and analyzed with NMRViewJ[42]. Backbone assignment of azurin and p53-DBD were obtained from reported work[43,44].

**Pull-down assay.** 200 μL purified strep-Azurin or strep-Azurin mutants at a concentration of 40 μM was incubated with 200 μl flag-p53-DBD, flag-p53-DBD mutants or flag-p53 full-length at 40 μM for 0.5 h at 4 °C in buffer containing 25 mM Tris-HCl and 150 mM NaCl at pH 8.0. The mixtures were incubated with 100 μl Anti-DYKDDDDK G1 Affinity Resin for 30 min at 4 °C. Then the Anti-DYKDDDDK G1 Affinity Resin were washed with TBS buffer containing 50 mM Tris-HCl and 150 mM NaCl at pH 7.4. The Resin were eluted with elution buffer containing DYKDDDDK peptide in TBS with concentration of 250 μg/ml, then resuspended with 100 μL of protein loading buffer and loaded into a 15% SDS-PAGE gel. Flag-p53-DBD or flag-p53 proteins were detected by western blotting using anti-DYKDDDDK-tag antibody (Proteintech, 66008-3-Ig, 1:3000 working dilution), and azurin was detected by anti-strep tag antibody (Abbkine, 8C12, ATUMR1201, 1:3000 working dilution).

**Ubiquitination assay.** GST-UBE1, UbcH5b, His-hCOP1, p53-His, Strep-azurin, and MSAC-Ub were over-expressed in *E. coli* BL21 (DE3) cell cultures and purified as above. The assays were performed in 50 μl reaction buffer containing 25 mM Tris (pH 7.5), 150 mM NaCl, 20 mM MgCl$_2$, 5 mM ATP, 0.5 μM purified GST-UBE1, 1 μM purified UBCH5b, 0.5 μM purified hCOP1, 2 μM p53-His, 1 μM fluorescently labeled ubiquitin (UbIR488) and 0, 4, 12, 36, 108 μM Strep-azurin, respectively. Ubiquitin was expressed and purified bearing a MSAC overhang at the N-terminus. The cysteine residue in the overhang was targeted FITC (MedMol, 75350-46-8) dye followed the reported method[40,45]. Control experiments of the ubiquitination assay were carried out in three conditions, (1) E1/E2/Ub/p53 in the

same concentration as above, (2) E1/E2/Ub/hCOP1, (3) E1/E2/Ub/hCOP1 supplied with wild type azurin at 108 μM. Reactions were incubated at 30 °C for 4 h, stopped by adding protein loading buffer, resolved by SDS-PAGE, and analyzed by using FLA5100 (Typhoon, Fuji, Japan). Each experiment was performed at least two individual times.

**Crystallization, data collection and structure determination**. Proteins for crystallization were purified and concentrated to 9 mg/mL. Crystallization was performed through hanging-drop vapor-diffusion method by mixing the proteins with an equal volume of reservoir solution at 18 °C. The best crystals for azurin-p53-DBD complex were obtained under the condition of 50 mM NaPi (pH 7.0), 19.5% PEG3350. Crystals were equilibrated in a cryoprotectant buffer containing 50% glycerol, 25 mM NaPi (pH 7.0), and 9.75% PEG3350. Eventually, a dataset with resolution at 2.1 Å was acquired at Shanghai Synchrotron Radiation Facility beamline BL17U. Azurin-p53-DBD complex crystallized in space group C121. X-ray diffraction data were collected and processed with HKL2000[46]. Four molecules in the asymmetric unit was indicated by Matthews coefficient calculation[47]. Molecular replacement was performed with PHASER[48] from PHENIX[49] package by utilizing the published monomeric DBD (PDB ID: 3KMD)[50] and azurin structure (PDB ID: 1E67)[51] as initial templates. Multi-round of structure refinement were performed with PHENIX[49] combined with COOT suit[52]. The statistics of our X-ray data collection and structural refinement are shown in the Table 1. The full view and the representative intermolecular interface of azurin-p53 DBD complex fitting into 2F0-Fc density map are shown in Fig. S8.

**Analytical ultracentrifugation (AUC)**. The AUC experiments were performed with Beckman Coulter XL-I analytical ultracentrifuge using two-channel centerpieces. The fusion protein p53_DBD92-292- linker (GGSGSGSLVPRGGSGGSGGS)-azurin was prepared in a solution of 25 mM Tris-HCl pH 8.0 and 150 mM NaCl. Data were collected via absorbance detection at 18 °C for proteins at a concentration of 0.75 mg/ml and rotor speed of 147,420 g. The SV-AUC data were globally analyzed using the SEDFIT program and fitted to a continuous c(s) distribution model to determine the molecular weight.

**Chemical crosslinking mass spectrometry**. Purified Flag-p53-DBD proteins and Strep-azurin were prepared in a solution containing 25 mM HEPES, pH 7.5 and 150 mM NaCl. Then 15 μL, 50 μM p53-DBD protein was mixed with 15 μL azurin protein at 50 μM on ice for 30 min. The protein mixtures were incubated with 37.5, 75, 150, and 300 mmol GA (Glutaraldehyde, Thermo Scientific, A1050022), BS2G (Bis(sulfosuccinimidyl) glutarate, Thermo Scientific, 21610) or BS3 (Bis(sulfosuccinimidyl)suberate, Thermo Scientific, 21580) for 30 min at 25 °C and then NH4HCO3 was added to 20 mM to quench the reaction. Products of the crosslinking reactions were analyzed by 15% SDS-PAGE and Western Blotting. For mass spectrometry, the destination fragment from the gel was cut from the SDS-PAGE. Samples were reduced, alkylated, and digested with trypsin. Digested peptides were separated from the solution, retained by lyophilization, and then separated by Q Exactive HF (Thermo Scientific) after adding 0.1% FA to the freeze-dried peptides. Data were analyzed by pLink Software[53].

**Cell-apoptosis analysis**. HEK293T Cells (105) were treated with 60 μM azurin and its mutants for 72 h. Cells were washed with PBS and resuspended in binding buffer. Then the cells were treated with a Propidium Iodide (PI)/Fluorescein Isothiocyanate (FITC)-Annexin V Staining Kit (TransGen, FA101) and subjected to flow cytometry analysis. A total of 106 cells were analyzed for each treatment, and each condition was tested in three independent times. FlowJo software was utilized to calculate the apoptotic cell rate. HEK293T cells were seeded into wells of a 6-well plate 1 day before transfection. Cells were transfected using the PEI transfection reagent under the conditions specified by the manufacturer. WT azurin plasmid or mutant azurin plasmids, and the empty plasmid were used to transfect the cells 48 and 72 h. The same transfection mix was prepared without the repair template as control. Cells were collected by centrifugation and disrupted in RIPA buffer containing 25 mM Tris-HCl (pH 8), 150 mM NaCl, 1% NP-40, 1% sodium deoxycholate, 0.1% SDS for 30 min. Protein contents were determined using NanoPhotometer®N60. In total, 40 μg of protein was separated on SDS–PAGE and electrophoretically transferred onto a PVDF membrane. Monoclonal antibodies raised against p21 (Cell signaling), p53 (Proteintech Group), Bax (Proteintech Group) were used for immunoblotting. Blots were also probed for β-actin by using monoclonal anti-actin antibody (Proteintech Group) as an internal control. Protein bands were visualized by using ECL reagents (Amersham Pharmacia).

**Statistics and reproducibility**. The statistical analysis including calculating of Pearson's correlations coefficient (P-value) was conducted with GraphPad Prism 7.0 software. The P-values were determined with one-way ANOVA followed by Dunnett's multiple comparisons test using PBS group as a control. Experiments related to statistics data were repeated at three independent preparations. Error bars indicate standard error of the mean (SEM) ($n = 3$, **$P < 0.001$, ***$P < 0.0001$). Source data are provided as a Source Data file.

**Reporting summary**. Further information on research design is available in the Nature Portfolio Reporting Summary linked to this article.

## Data availability

Atomic coordinate for the crystal structure has been deposited in the Protein Data Bank under accession number 7YGI. Source data for the cell apoptosis assay is included in the Supplementary Data 1. Raw images of the related SDS-PAGE gels and western blots are provided in Supplementary Figs. 10–13. Other data are presented in this article and its Supplementary Information files, or can be obtained from the corresponding author upon request.

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

## Acknowledgements

This work was supported by the Green Bio-producing of National Key R&D Program of China (grant no. 2021YFC2100100) and the Open Project Funding of the State Key Laboratory of Biocatalysis and Enzyme Engineering (grant no. SKLBEE2020014). The authors acknowledge Yixin Cui from St.Jude Children's research hospital, Bo Wang from Xiamen University, Chengdong Huang from University of Science and Technology of China, Yajun Jiang from Nanking University, for critically reading and helpful discussion relating to this project. The authors also grateful to the staff of NCPSS at the Shanghai Synchrotron Radiation Facility for data collection. NMR spectra were acquired at the National Center for Magnetic Resonance in Wuhan.

## Author contributions

Q.X. conceived the project. J.H. performed the pull-down assay and apoptosis assay. W.J. prepared all the samples for NMR experiments, performed the NMR experiments and analysis the data. J.H., J.Z. and D.S. prepared all the protein samples, conducted the crystal screening and X-ray data collection, AUC experiments, mass spectrometry experiments and ubiquitination assay. X.C., X.Y. and W.Z. assisted to make all the plasmids. L.M., Z.L., and Q.X. designed the experiments and interpreted the data. J.H., W.J., J.Z. and Q.X. wrote the paper. All authors reviewed and approved the manuscript.

## Competing interests

The authors declare no competing interests.
