## [Peer Review File · Communications Biology]

Reviewers' comments:

Reviewer #1 (Remarks to the Author):

The paper describes structural and functional studies detailing the molecular interactions of the bacterial effector protein azurin with tumor suppressor p53. Modulation of p53 plays an important role in cancer therapy so this work has relevance to the wider community.

The authors report NMR studies demonstrating the regions of binding on the respective proteins. These studies indicate weak binding interactions. Key residues involved in the binding interactions are identified. The NMR structural data was used to design fusion proteins for crystallography leading to the solution of a 2.2 Å X-ray structure of azurin-p53 fusion protein. The interface of azurin with p53 is described in detail. Understanding of the protein-protein interface allowed the investigation of the role of individual surface amino acids allowed the identification of azurin mutants with improved binding affinity, The authors provide data that shows the azurin mutants have downstream cellular effects mediated by stabilised p53.

The methods are described in adequate detail. However, the presentation of the results is not always as clear as it could be. In particular, the supporting figures showing the 2D NMR data are of a low resolution and as the chemical shift changes are small, they are not easy to see in the figures despite the labelling. The figures of the X-ray structure complex are well presented. A figure showing the binding of each protein displayed with a surfaces colored by polarity and hydrophobicity with the complementary proteins amino acids would greatly help in the understanding of the nature of the interface. Although H-bond interactions are identified, these may not have a large contribution to the binding energy, so surface complementarity for shape and properties should be illustrated and discussed.

Some of the blots used in the figures are not very high quality or look overexposed. In some cases, it is difficult to understand the conclusions drawn from the data shown in the figures, e.g. fig 2b is intended to show decreased stability of the complex, but the authors do not state clearly why the changes observed lead to this conclusion. Fig 3d in particular has poor quality images for the blots. The changes to ubiquitination are evident as described but the blots are not compelling. The changes described in Fig 4c are more evident and these images are of a higher quality. Overall, the data shown supports the authors discussion, but the figures could be improved.

The paper offers an interesting approach to p53 stabilisation that may ultimately have a therapeutic endpoint.

Reviewer #2 (Remarks to the Author):

Hu et al. reported a transient and dynamic interaction between the bacterial toxin azurin and the tumor suppressor p53. The authors first used solution NMR to capture the weak interaction between azurin and p53, then mapped the interaction domain to the DNA binding domain (DBD) of p53. They then solved the crystal structure of azurin in complex with p53 DBD. Interestingly, the crystal structure shows one azurin molecule binds to two p53 DBDs, in a way similar to how the anti-apoptotic protein Bcl-xL binds to p53. The azurin binding interface on p53 partially overlaps with COP1 footprint on p53. Mutations on the interface disrupts azurin: p53 interaction and restores COPI-mediated p53 ubiquitination in vitro. The authors also mutated interfacial residues in the hope to enhance azurin:p53 interaction and decreases p53 ubiquitination. This set of mutants interact with p53 with mildly enhanced affinity, but drastically decrease p53 ubiquitination. Furthermore, when added to cells, these stabilizing mutants lead to more cell apoptosis, demonstrating their potential in anticancer treatment.

The authors used a combination of structural biology and cellular biology to address an important issue in cancer biology. The logic is clear and the experimental design is solid. However, the author

should address the following issues:

Major concerns:

1. The authors reported the crystal structure resolution as 2.2 Å. However, in Table S1, the R-merge, R-meas, R-pim, and CC1/2 values are unusual for a 2.2 Å structure. Furthermore, the Rwork/Rfree values of 26.5%/33.5% are too high for a 2.2 Å structure. The abnormal statistics suggest the structure does not reach the resolution, and/or a large region of the structure does not fit well with the electron densities. The author should re-process the diffraction data and re-refine the structure.
2. Structure representation in Figures 1 and 2 is poor. The tube representation makes it difficult to discern structural features. In some cases it is almost impossible to tell mainchain carbonyl oxygen from the sidechain hydroxyl group. The authors should consider other types of representation or software to clearly present the structural features. The colors of different azurin regions are too close to each other. More contrasting colors should be used.

Other concerns:

1. Proteins are generated in this manuscript, but in most cases they are detected by Western Blot. Representative size exclusion chromatography profiles and/or SDS-PAGE of azurin and p53 should be included to show homogeneity and purity of the proteins.
2. Though the domain boundaries are marked in Figure 1a, in the main text and Methods section the p53 constructs boundaries are not clear. Please specify the starting and ending residues of p53-NT, -CT, and -DBD either in the main text or in the methods.
3. In all Western Blot, please include molecular markers that are closest to the proteins detected, as the authors did in Supplementary Figure 7. Please also state the number of replicates for each Western Blot, either in the legends or in the Methods.
4. The reported crystal structure comes from a p53-DBD-GSlinier-azurin fusion protein. To preclude any fusion-induced artificial interaction, please briefly describe the linker positions in the main text and mark the two linkers in Supplementary Figure 3a.
5. Line 149-152, the authors stated that the SV-AUC experiment showed the fusion protein is in an equilibrium of monomer and dimer in solution. The claim is dubious, as the monomeric 39.2 kDa species is predominant. Could the authors elaborate more? Also, the main text mentions the SV-AUC experiments used the same protein concentration as crystallization, which is consistent with the 7.9 mg/mL in Supplementary Figure 3b legend. However, in the Methods section, line 429, the concentration is 0.75 mg/mL. Please clarify.
6. Line 159, is the 913 Å² buried area on one molecule or the sum of buried areas on both azurin and p53?
7. In Figure 2a Leu 188 points away from the azurin molecule; yet in line 161 the authors wrote Leu188 interacts with a hydrophobic pocket of azurin. Could the authors elaborate on the role of Leu188?
8. Figure 3c: at the higher azurin concentration, the intensity of the p53 bands in both 2h and 4h incubation are less than those with no azurin or lower azurin concentration. This weakens the argument that with higher azurin concentration reduces p53 ubiquitination. Could the authors elaborate on this? In addition, did the authors observe a p53-Ub band in the anti-p53 blot?
9. Supplementary Figure 4: please provide the UniProt identifiers of the 20 azurin sequences.
10. The authors hypothesize that azurin competes with Bcl-xL for p53 dimer binding. Could the

authors compare the binding affinities of azurin and Bcl-xL to p53?

11. From the text my understanding is azurin is Strep-tagged. The method also mentioned azurin is purified with Strep-Tactin columns. However, the cloning section describes azurin as His-tagged. Please clarify. Also, is it pET-15d or pET-15b vector?

12. Cloning and purification of GST-DBD (Figure 2b) are not described. Actually in the Methods section the pull-down assay is said to be between Flag-tagged p53-DBD, please clarify.

13. Line 344 and 399. MSAC-Ub, please spell out MSAC and cite proper references or manufacture protocols for FITC labeling of MSAC-Ub.

14. Line 400-401, there is no mentioning of hCOP1 autoubiquitination in the main text, please clarify.

15. Line 413, the space group is stated as P1212 here. In Table S1 it is C 1 2 1. Please clarify.

16. Line 414, please cite the reference for HKL2000.

Reviewer #3 (Remarks to the Author):

This paper studies the structural basis of the interaction between a bacterial protein azurin and p53, using both solution NMR and X-ray. Based on the high-resolution structure of the protein complex, affinity enhancing mutations in azurin were designed and shown to enhance binding, inhibit p53 ubiquitination and enhance apoptosis through p53 stabilization. Overall, this work is very well done and has important application for improved azurin derived peptide drugs for cancer treatment. I highly recommend its publications. I only have a few minor concerns:

1. P53 is a tetramer in cell. The author may want to discuss the implication of DBD dimer interaction with azurin in this context.
2. Fig. 5 legend: panel descriptions do not correspond to the figure contents. Please correct.
3. Line 274, "apoptosis senescence": there should be a comma between these two words.
4. Line 275, "p53 inactivation": I rather use "p53 mutations" because there are gain-of-function p53 mutations.
5. Line 35, recommend rewording "official p28 region", for example, to "p28 region, which corresponds to a therapeutic peptide".
6. Line 40, "derivates" should be changed to "derivative"
7. Line 321-322: The first sentence of the paragraph sounds a little awkward. Reword to "Currently, major efforts are devoted to develop drugs to restore p53 stability and activity".

*Reviewers' comments:*

*Reviewer #1 (Remarks to the Author):*

*The paper describes structural and functional studies detailing the molecular interactions of*
*the bacterial effector protein azurin with tumor suppressor p53. Modulation of p53 plays an*
*important role in cancer therapy so this work has relevance to the wider community.*

*The authors report NMR studies demonstrating the regions of binding on the respective*
*proteins. These studies indicate weak binding interactions. Key residues involved in the binding*
*interactions are identified. The NMR structural data was used to design fusion proteins for*
*crystallography leading to the solution of a 2.2 Å X-ray structure of azurin-p53 fusion protein.*

*The interface of azurin with p53 is described in detail. Understanding of the protein-protein*
*interface allowed the investigation of the role of individual surface amino acids allowed the*
*identification of azurin mutants with improved binding affinity, The authors provide data that*
*shows the azurin mutants have downstream cellular effects mediated by stabilised p53.*

*The methods are described in adequate detail. However, the presentation of the results is not*
*always as clear as it could be. In particular, the supporting figures showing the 2D NMR data*
*are of a low resolution and as the chemical shift changes are small, they are not easy to see in*
*the figures despite the labelling. The figures of the X-ray structure complex are well presented.*

*A figure showing the binding of each protein displayed with a surfaces colored by polarity and*
*hydrophobicity with the complementary proteins amino acids would greatly help in the*
*understanding of the nature of the interface. Although H-bond interactions are identified, these*
*may not have a large contribution to the binding energy, so surface complementarity for shape*
*and properties should be illustrated and discussed.*

*Some of the blots used in the figures are not very high quality or look overexposed. In some*
*cases, it is difficult to understand the conclusions drawn from the data shown in the figures,*
*e.g. fig 2b is intended to show decreased stability of the complex, but the authors do not state*

*clearly why the changes observed lead to this conclusion. Fig 3d in particular has poor quality*
*images for the blots. The changes to ubiquitination are evident as described but the blots are*
*not compelling. The changes described in Fig 4c are more evident and these images are of a*
*higher quality. Overall, the data shown supports the authors discussion, but the figures could*
*be improved.*

*The paper offers an interesting approach to p53 stabilisation that may ultimately have a*
*therapeutic endpoint.*

*Reviewer #2 (Remarks to the Author):*

*Hu et al. reported a transient and dynamic interaction between the bacterial toxin azurin and*
*the tumor suppressor p53. The authors first used solution NMR to capture the weak interaction*
*between azurin and p53, then mapped the interaction domain to the DNA binding domain*
*(DBD) of p53. They then solved the crystal structure of azurin in complex with p53 DBD.*
*Interestingly, the crystal structure shows one azurin molecule binds to two p53 DBDs, in a way*
*similar to how the anti-apoptotic protein Bcl-xL binds to p53. The azurin binding interface on*
*p53 partially overlaps with COPI footprint on p53. Mutations on the interface disrupts azurin:*
*p53 interaction and restores COPI-mediated p53 ubiquitination in vitro. The authors also*
*mutated interfacial residues in the hope to enhance azurin:p53 interaction and decreases p53*
*ubiquitination. This set of mutants interact with p53 with mildly enhanced affinity, but*
*drastically decrease p53 ubiquitination. Furthermore, when added to cells,*
*these stabilizing mutants lead to more cell apoptosis, demonstrating their potential in*
*anticancer treatment.*

*The authors used a combination of structural biology and cellular biology to address an*
*important issue in cancer biology. The logic is clear and the experimental design is solid.*

*However, the author should address the following issues:*

*Major concerns:*

*1. The authors reported the crystal structure resolution as 2.2 Å. However, in Table S1, the R-*
*merge, R-meas, R-pim, and CCI/2 values are unusual for a 2.2 Å structure. Furthermore, the*
*Rwork/Rfree values of 26.5%/33.5% are too high for a 2.2 Å structure. The abnormal statistics*
*suggest the structure does not reach the resolution, and/or a large region of the structure does*
*not fit well with the electron densities. The author should re-process the diffraction data and*
*re-refine the structure.*

*2. Structure representation in Figures 1 and 2 is poor. The tube representation makes it difficult*
*to discern structural features. In some cases it is almost impossible to tell mainchain carbonyl*
*oxygen from the sidechain hydroxyl group. The authors should consider other types of*
*representation or software to clearly present the structural features. The colors of different*
*azurin regions are too close to each other. More contrasting colors should be used.*

*Other concerns:*

*1. Proteins are generated in this manuscript, but in most cases they are detected by Western*
*Blot. Representative size exclusion chromatography profiles and/or SDS-PAGE of azurin and*
*p53 should be included to show homogeneity and purity of the proteins.*

*2. Though the domain boundaries are marked in Figure 1a, in the main text and Methods*
*section the p53 constructs boundaries are not clear. Please specify the starting and ending*
*residues of p53-NT, -CT, and -DBD either in the main text or in the methods.*

*3. In all Western Blot, please include molecular markers that are closest to the proteins*
*detected, as the authors did in Supplementary Figure 7. Please also state the number of*
*replicates for each Western Blot, either in the legends or in the Methods.*

4. The reported crystal structure comes from a p53-DBD-GSlinier-azurin fusion protein. To
preclude any fusion-induced artificial interaction, please briefly describe the linker positions
in the main text and mark the two linkers in Supplementary Figure 3a.

5. Line 149-152, the authors stated that the SV-AUC experiment showed the fusion protein is
in an equilibrium of monomer and dimer in solution. The claim is dubious, as the monomeric
39.2 kDa species is predominant. Could the authors elaborate more? Also, the main text
mentions the SV-AUC experiments used the same protein concentration as crystallization,
which is consistent with the 7.9 mg/mL in Supplementary Figure 3b legend. However, in the
Methods section, line 429, the concentration is 0.75 mg/mL. Please clarify.

6. Line 159, is the 913 Å² buried area on one molecule or the sum of buried areas on both
azurin and p53?

7. In Figure 2a Leu 188 points away from the azurin molecule; yet in line 161 the authors wrote
Leu188 interacts with a hydrophobic pocket of azurin. Could the authors elaborate on the role
of Leu188?

8. Figure 3c: at the higher azurin concentration, the intensity of the p53 bands in both 2h and
4h incubation are less than those with no azurin or lower azurin concentration. This weakens
the argument that with higher azurin concentration reduces p53 ubiquitination. Could the
authors elaborate on this? In addition, did the authors observe a p53-Ub band in the anti-p53
blot?

9. Supplementary Figure 4: please provide the UniProt identifiers of the 20 azurin sequences.

10. The authors hypothesize that azurin competes with Bcl-xL for p53 dimer binding. Could
the authors compare the binding affinities of azurin and Bcl-xL to p53?

11. From the text my understanding is azurin is Strep-tagged. The method also mentioned
azurin is purified with Strep-Tactin columns. However, the cloning section describes azurin as
His-tagged. Please clarify. Also, is it pET-15d or pET-15b vector?

*12. Cloning and purification of GST-DBD (Figure 2b) are not described. Actually in the*
*Methods section the pull-down assay is said to be between Flag-tagged p53-DBD, please*
*clarify.*

*13. Line 344 and 399. MSAC-Ub, please spell out MSAC and cite proper references or*
*manufacture protocols for FITC labeling of MSAC-Ub.*

*14. Line 400-401, there is no mentioning of hCOP1 autoubiquitination in the main text, please*
*clarify.*

*15. Line 413, the space group is stated as P1212 here. In Table S1 it is C 1 2 1. Please clarify.*

*16. Line 414, please cite the reference for HKL2000.*

*Reviewer #3 (Remarks to the Author):*

*This paper studies the structural basis of the interaction between a bacterial protein azurin*
*and p53, using both solution NMR and X-ray. Based on the high-resolution structure of the*
*protein complex, affinity enhancing mutations in azurin were designed and shown to enhance*
*binding, inhibit p53 ubiquitination and enhance apoptosis through p53 stabilization. Overall,*
*this work is very well done and has important application for improved azurin derived peptide*
*drugs for cancer treatment. I highly recommend its publications. I only have a few minor*
*concerns:*

*1. P53 is a tetramer in cell. The author may want to discuss the implication of DBD dimer*
*interaction with azurin in this context.*

*2. Fig. 5 legend: panel descriptions do not correspond to the figure contents. Please correct.*

*3. Line 274, "apoptosis senescence": there should be a comma between these two words.*

4. Line 275, “p53 inactivation”: I rather use “p53 mutations” because there are gain-of-
function p53 mutations.

5. Line 35, recommend rewording “official p28 region”, for example, to “p28 region, which
corresponds to a therapeutic peptide”.

6. Line 40, “derivates” should be changed to “derivative”

7. Line 321-322: The first sentence of the paragraph sounds a little awkward. Reword to
“Currently, major efforts are devoted to develop drugs to restore p53 stability and activity”.

Response to referees.

Reviewer #1 (Remarks to Author)

The paper describes structural and functional studies detailing the molecular interactions of the
bacterial effector protein azurin with tumor suppressor p53. Modulation of p53 plays an
important role in cancer therapy so this work has relevance to the wider community.

The authors report NMR studies demonstrating the regions of binding on the respective
proteins. These studies indicate weak binding interactions. Key residues involved in the
binding interactions are identified. The NMR structural data was used to design fusion proteins
for crystallography leading to the solution of a 2.2 Å X-ray structure of azurin-p53 fusion
protein. The interface of azurin with p53 is described in detail. Understanding of the protein-
protein interface allowed the investigation of the role of individual surface amino acids allowed
the identification of azurin mutants with improved binding affinity, The authors provide data
that shows the azurin mutants have downstream cellular effects mediated by stabilised p53.

Re: Thank you very much for all your comments

The methods are described in adequate detail. However, the presentation of the results is not
always as clear as it could be. In particular, the supporting figures showing the 2D NMR data
are of a low resolution and as the chemical shift changes are small, they are not easy to see in
the figures despite the labelling.

Re: We modified the supplementary figure 1 and highlighted the representative peaks which
are perturbed on the NMR spectrum.

The figures of the X-ray structure complex are well presented. A figure showing the binding
of each protein displayed with a surface colored by polarity and hydrophobicity with the
complementary proteins amino acids would greatly help in the understanding of the nature of
the interface.

Re: Thanks for your suggestion. A figure of p53-DBD and azurin displayed as surfaces and
highlighted charge as well as hydrophobicity has been added as supplementary figure 5. The
solvent-exposed surface rendering of the complex was displayed in the Yellow-Red-Blue color
scheme.

Although H-bond interactions are identified, these may not have a large contribution to the
binding energy, so surface complementarity for shape and properties should be illustrated and
discussed.

Re: We made a figure as supplementary fig. 5 to elaborate the surface complementarity and
revised the text and added the discussion of surface complementarity in the structure analysis
part. The sentence we added in the main text is “Dimerization of p53-DBD forms a triangle
groove, which is highly complementary to the triangle shape of azurin (Supplementary
Fig. 5b). This indicates the contribution of van der Waals interactions in the complex
formation.”

Some of the blots used in the figures are not very high quality or look overexposed. In some
cases, it is difficult to understand the conclusions drawn from the data shown in the figures,
e.g. fig 2b is intended to show decreased stability of the complex, but the authors do not state
clearly why the changes observed lead to this conclusion.

Re: Thanks very much for your comments and sorry for the confusion. We have already revised
figure 2b as well as figure 4b and made a statement in the figure legend as “This assay was
performed with anti-Flag beads which specifically binds to Flag-DBD or its mutants to
pull down strep-azurin or its mutants in solution, and the elution was blotted with anti-
flag and anti-strep antibodies, respectively. The intensity of the blot of azurin
represents the relative affinity of the azurin and p53-DBD.”

Fig 3d in particular has poor-quality images for the blots. The changes to ubiquitination are
evident as described but the blots are not compelling. The changes described in Fig 4c are more
evident and these images are of a higher quality.

Re: The suppression of wild-type azurin on p53 ubiquitination in our ubiquitination assay is
obvious but not dramatic, and the suppression of affinity-decreasing azurin mutants on p53
ubiquitination is barely seen, which exactly backup our conclusion that azurin interferes p53
ubiquitination. On the contrary, when it comes to affinity-enhancing mutants, the effect of
azurin mutants on decreasing p53 ubiquitination is much more significant as in Fig 4c.

Overall, the data shown supports the author’s discussion, but the figures could be improved.
The paper offers an interesting approach to p53 stabilization that may ultimately have a
therapeutic endpoint.

Re: Thanks very much for your significant suggestions.

Reviewer #2 (Remarks to the Author):

Hu et al. reported a transient and dynamic interaction between the bacterial toxin azurin and
the tumor suppressor p53. The authors first used solution NMR to capture the weak interaction
between azurin and p53, then mapped the interaction domain to the DNA binding domain
(DBD) of p53. They then solved the crystal structure of azurin in complex with p53 DBD.
Interestingly, the crystal structure shows one azurin molecule binds to two p53 DBDs, in a way
similar to how the anti-apoptotic protein Bcl-xL binds to p53. The azurin binding interface on
p53 partially overlaps with COP1 footprint on p53. Mutations on the interface disrupts azurin:
p53 interaction and restores COPI-mediated p53 ubiquitination in vitro. The authors also
mutated interfacial residues in the hope to enhance azurin:p53 interaction and decreases p53
ubiquitination. This set of mutants interact with p53 with mildly enhanced affinity, but
drastically decrease p53 ubiquitination. Furthermore, when added to cells, these stabilizing
mutants lead to more cell apoptosis, demonstrating their potential in anticancer treatment.

Re: Thank you very much for your comments.

The authors used a combination of structural biology and cellular biology to address an
important issue in cancer biology. The logic is clear and the experimental design is solid.

However, the author should address the following issues:

Major concerns:

1. The authors reported the crystal structure resolution as 2.2 Å. However, in Table S1, the R-
merge, R-meas, R-pim, and CC1/2 values are unusual for a 2.2 Å structure. Furthermore, the
Rwork/Rfree values of 26.5%/33.5% are too high for a 2.2 Å structure. The abnormal statistics
suggest the structure does not reach the resolution, and/or a large region of the structure does

not fit well with the electron densities. The author should re-process the diffraction data and
re-refine the structure.

Re: Thanks very much for this comment. We carefully reprocessed the diffraction data and re-
refined the structure, updated the statistics table of the crystal structure, and deposited our new
structure to the PDB server. The Rmerge/Rmeas values are 13.2%/14.7%, and the Rwork/Rfree
values are 21.1% /26.5%. To further validate the structure, we highlighted the interface and
overall structure fitting into the density map in Supplementary Fig 7. It shows that the structure
fits very well with the electron densities.

2. Structure representation in Figures 1 and 2 is poor. The tube representation makes it difficult
to discern structural features. In some cases it is almost impossible to tell mainchain carbonyl
oxygen from the sidechain hydroxyl group. The authors should consider other types of
representation or software to clearly present the structural features. The colors of different
azurin regions are too close to each other. More contrasting colors should be used.

Re: We have already changed all the fatty tube representations of our structure to normal
cartoon representations. The colors of different azurin regions in Fig.1 have been changed to
dark cyan for p28 and marine for CT-p26.

Other concerns:

1. Proteins are generated in this manuscript, but in most cases they are detected by Western
Blot. Representative size exclusion chromatography profiles and/or SDS-PAGE of azurin and
p53 should be included to show homogeneity and purity of the proteins.

Re: We added a series of images in supplementary fig.1 to include the representative size
exclusion chromatography profiles as well as SDS-Page for p53, p53-DBD, and azurin. These
results show that all the proteins are highly homogenous and pure.

2. Though the domain boundaries are marked in Figure 1a, in the main text and Methods section
the p53 constructs boundaries are not clear. Please specify the starting and ending residues of
p53-NT, -CT, and -DBD either in the main text or in the methods.

Re: We have already updated the domain boundaries of the p53 protein in the main text.

3. In all Western Blot, please include molecular markers that are closest to the proteins detected,
as the authors did in Supplementary Figure 7. Please also state the number of replicates for
each Western Blot, either in the legends or in the Methods.

Re: The molecular markers have been added to all the western blot figures and we also made
the statement of the replicate times of each experiment in the Methods part (line 414).

4. The reported crystal structure comes from a p53-DBD-GSlinier-azurin fusion protein. To
preclude any fusion-induced artificial interaction, please briefly describe the linker positions
in the main text and mark the two linkers in Supplementary Figure 3a.

Re: We remade supplementary figure 3 and added a scheme and structural representation of
our fusion construct. In addition to this, we also added a sentence to describe the fusion
construct in our main text , “The highest quality crystals suitable for structural

determination were given by a construct of fusing azurin to the C-terminal of p53-DBD
with a glycine-serine repeats linker (GGSGSGSLVPRGGSGGSGGS).”

5. Line 149-152, the authors stated that the SV-AUC experiment showed the fusion protein is
in an equilibrium of monomer and dimer in solution. The claim is dubious, as the monomeric
39.2 kDa species is predominant. Could the authors elaborate more? Also, the main text
mentions the SV-AUC experiments used the same protein concentration as crystallization,
which is consistent with the 7.9 mg/mL in Supplementary Figure 3b legend. However, in the
Methods section, line 429, the concentration is 0.75 mg/mL. Please clarify.

Re: Thanks for the comment. Supplementary Figure 3 in our manuscript is in very low
resolution, which makes the dimer peak ambiguous. We have revised this figure and made it
in high resolution. Actually, we repeated this experiment two times. The dimer peak was
presented in both replicants. We also try to use higher concentrations, but the UV absorbance
from our UV detector on SV-AUC always overflowed, and it didn't give us a rational result.
In addition, the dimerization of p53_DBBD (in the absence of DNA) has been characterized as
a very weak interaction by multiple published works, which explains our SV-AUC data that
the monomeric fraction is predominant compared to the dimer fraction at a relatively low
protein concentration.

We double-checked our experiments carefully. The sample concentration for SV-AUC
experiment is 0.75 mg/mL as the method part. We did run an SV-AUC experiment with 7.9
318 mg/mL sample, the UV absorbance overflowed as the concentration is too high. We have
319 already corrected our manuscript.

6. Line 159, is the 913 Å² buried area on one molecule or the sum of buried areas on both
azurin and p53?

Re: The 913 Å² buried area is the sum of the buried area on both azurin and p53 sides.

7. In Figure 2a Leu 188 points away from the azurin molecule; yet in line 161 the authors wrote
Leu188 interacts with a hydrophobic pocket of azurin. Could the authors elaborate on the role
of Leu188?

Re: Thanks for the comment. We double-checked our re-refined structure and confirmed that
the Leu 188 points to azurin. The figure. 2a was revised. Actually, Leu 188 is very near to
azurin molecular, the distance between Leu 188 methyl groups to azurin is less than 4 Å, which
validates the hydrophobic interaction between these residues.

8. Figure 3c: at the higher azurin concentration, the intensity of the p53 bands in both 2h and
4h incubation are less than those with no azurin or lower azurin concentration. This weakens
the argument that with higher azurin concentration reduces p53 ubiquitination. Could the
authors elaborate on this? In addition, did the authors observe a p53-Ub band in the anti-p53
blot?

Re: Thanks for this comment. We did notice that the intensity of the p53 band is a little bit less
compared to the mixture without azurin due to sample loading. As p53 ubiquitination level is
dramatically different, we think that it is acceptable to back up our result. Also, we initially did
use anti-p53 blot to perform the ubiquitination assay. As the p53-Ub was much less than p53

in the mixture, we can only see a long-blurred blot with an anti-p53 antibody. So, we changed
to anti-Ub blot.

9. Supplementary Figure 4: please provide the UniProt identifiers of the 20 azurin sequences.

Re: We updated the figure legend and added all the uniport ID of azurin sequences.

10. The authors hypothesize that azurin competes with Bcl-xL for p53 dimer binding. Could
the authors compare the binding affinities of azurin and Bcl-xL to p53?

Re: We could have a conclusion that the binding affinities of azurin to p53 is weaker than Bcl-
xL to p53 in vitro based on our NMR titration experiments and the reported data. As azurin is
secreted from bacteria to cell, the concentration of azurin could be much higher than Bcl-xL.
Based on the concentration effect, we hypothesize that azurin could compete with Bcl-xL for
p53 binding in cell.

11. From the text my understanding is azurin is Strep-tagged. The method also mentioned
azurin is purified with Strep-Tactin columns. However, the cloning section describes azurin as
His-tagged. Please clarify. Also, is it pET-15d or pET-15b vector?

Re: We carefully checked our manuscript and our experiment notes, and confirmed that the
construct for azurin is strep tagged. We have already revised this typo. And the vector we used
here is pET-15d.

12. Cloning and purification of GST-DBD (Figure 2b) are not described. Actually in the
Methods section, the pull-down assay is said to be between Flag-tagged p53-DBD, please
clarify.

Re: Sorry for the typo. The construct of DBD used for pull-down assay is always flag-tagged.
I have already revised Figure 2b.

13. Line 344 and 399. MSAC-Ub, please spell out MSAC and cite proper references or
manufacture protocols for FITC labeling of MSAC-Ub.

Re: Citations of this method have been added and we added a sentence to describe the MSAC-
Ub construct in the Method part. “MSAC-Ub (Four residues Met, Ser, Ala and Cys
appended at the N-terminus of ubiquitin).”.

14. Line 400-401, there is no mention of hCOP1 autoubiquitination in the main text, please
clarify.

Re: The word “auto-ubiquitination” we used was to describe the control experiment, which is
not precise and misleading. We have changed this sentence and added it to the Methods part.
“Control experiments of the ubiquitination assay were carried out in three conditions, 1)
E1/E2/Ub/p53 in the same concentration as above, 2) E1/E2/Ub/hCOP1, 3) E1/E2/Ub/hCOP1
supplied with wild type azurin at 108 μ M.”

15. Line 413, the space group is stated as P1212 here. In Table S1 it is C 1 2 1. Please clarify.

Re: We double-checked our data, and the space group is C 1 2 1. We have already revised it in
the text.

16. Line 414, please cite the reference for HKL2000.

Re: We have cited the original paper of HKL2000.

Reviewer #3 (Remarks to the Author):

This paper studies the structural basis of the interaction between a bacterial protein azurin and
p53, using both solution NMR and X-ray. Based on the high-resolution structure of the protein
complex, affinity-enhancing mutations in azurin were designed and shown to enhance binding,
inhibit p53 ubiquitination and enhance apoptosis through p53 stabilization. Overall, this work
is very well done and has important applications for improved azurin-derived peptide drugs for
cancer treatment. I highly recommend its publications. I only have a few minor concerns:

1. P53 is a tetramer in cell. The author may want to discuss the implication of DBD dimer
interaction with azurin in this context.

Re: Thanks for this suggestion. We added a sentence in the discussion part line 307. “Stable
tetramerization of p53 in cell would enhance the interaction for azurin, which indicates the
effect of azurin induced p53 stabilization would be amplified in cell comparing to our in vitro
ubiquitination assay.”

2. Fig. 5 legend: panel descriptions do not correspond to the figure contents. Please correct.

Re: Thanks, we have already corrected it.

3. Line 274, “apoptosis senescence”: there should be a comma between these two words.

Re: We have already revised it.

4. Line 275, “p53 inactivation”: I rather use “p53 mutations” because there are gain-of-function

p53 mutations.

Re: We have already substituted “p53 inactivation” with “p53 mutations”.

5. Line 35, recommend rewording “official p28 region”, for example, to “p28 region, which

corresponds to a therapeutic peptide”.

Re: We have revised this sentence as recommended.

6. Line 40, “derivates” should be changed to “derivative”

Re: “Derivates” have been changed to derivatives.

7. Line 321-322: The first sentence of the paragraph sounds a little awkward. Reword to

“Currently, major efforts are devoted to developing drugs to restore p53 stability and activity”.

Re: Thanks very much. I have already changed this sentence in the text.

Reviewers' comments:

Reviewer #1 (Remarks to the Author):

The authors have responded to all of the criticisms of the paper. Presentation of figures has been improved greatly. Additional experimental details and data has been added. In particular the figures in the supporting information have been added to with informative data. The resulting paper is significantly improved and is now recommended for publication.

Reviewer #2 (Remarks to the Author):

The authors addressed most of my concerns. I am in general satisfied with the authors' responses, but would like the authors to clarify the two points below.

1. Crystallography statistics are much improved after re-processing and re-refinement. The only part that is less satisfactory is completeness: in Table S1, the completeness is 84.6% for all resolution shells, which is unusual for crystallographic datasets of this resolution. Do the authors have an explanation for the low completeness?

2. Is the linker visible in the electron density map? Can the authors trace a fusion protein from N- to C- terminus without any ambiguity? The distance between the C-terminus of DBD_A and the N-terminus of azurin_A seems quite long (Supplementary Figure 3). Is the linker length sufficient to cover that distance? The C-terminus of DBD_A (top AU) is closer to the N-terminus of azurin_A in the bottom AU, is it likely that these two molecules actually belong to the same fusion protein?

*Reviewers' comments:*

*Reviewer #2 (Remarks to the Author):*

*The authors addressed most of my concerns. I am in general satisfied with the authors'*
*responses, but would like the authors to clarify the two points below.*

*1. Crystallography statistics are much improved after re-processing and re-refinement. The*
*only part that is less satisfactory is completeness: in Table S1, the completeness is 84.6% for*
*all resolution shells, which is unusual for crystallographic datasets of this resolution. Do the*
*authors have an explanation for the low completeness?*

*2. Is the linker visible in the electron density map? Can the authors trace a fusion protein from*
*N- to C- terminus without any ambiguity? The distance between the C-terminus of DBD_A and*
*the N-terminus of azurin_A seems quite long (Supplementary Figure 3). Is the linker length*
*sufficient to cover that distance? The C-terminus of DBD_A (top AU) is closer to the N-*
*terminus of azurin_A in the bottom AU, is it likely that these two molecules actually belong to*
*the same fusion protein?*

Response to referees.

Reviewer #2 (Remarks to Author)

The authors addressed most of my concerns. I am in general satisfied with the authors'
responses, but would like the authors to clarify the two points below.

Re: Thank you very much for all your suggestions.

1. Crystallography statistics are much improved after re-processing and re-refinement. The
only part that is less satisfactory is completeness: in Table S1, the completeness is 84.6% for
all resolution shells, which is unusual for crystallographic datasets of this resolution. Do the
authors have an explanation for the low completeness?

Re: We quickly reprocess the data and re-refined the structure. The completeness now is 96.18%
(99.22%) and the other parameters remain similar to the last version as presented in the updated
supplementary table 1. We have already redeposited our new structure and processed data into
PDB.

2. Is the linker visible in the electron density map? Can the authors trace a fusion protein from
N- to C- terminus without any ambiguity? The distance between the C-terminus of DBD_A
and the N-terminus of azurin_A seems quite long (Supplementary Figure 3). Is the linker length
sufficient to cover that distance? The C-terminus of DBD_A (top AU) is closer to the N-
terminus of azurin_A in the bottom AU, is it likely that these two molecules actually belong to
the same fusion protein?

Re: Thank you very much for this comment. As the linker part is full of Gly and Ser residues
in highly dynamic and flexible properties, the linker part and about 6 residues (4 residues
located at the C-terminus of DBD and 2 residues at the N-terminus of azurin) connected to the
linker part are invisible. This feature makes us disable tracing the exact linker position.
The distance between the visible C-terminus of DBD_A and the visible N-terminus of azurin_A
is 58 Å. The GS-linker residues plus the invisible and flexible terminal residues are about 26
residues. Theoretically, the average length of a residue backbone in a disordered peptide is
about 3.6 Å, so we think the linker length is sufficient to cover the distance. Although the C-
terminus of DBD_A (top AU) is closer to the N-terminus of azurin_A in the bottom AU while
there is no direct interaction between these two molecules, so we don't think these two
molecules belong to the same fusion construct.